# Analysis and Comprehensive Evaluation of Water Use Efficiency in China

**Wenge Zhang [1],\*, Xianzeng Du [2], Anqi Huang [2] and Huijuan Yin [1]**

[1] Yellow River Institute of Hydraulic Research, Yellow River Conservancy Commission, Zhengzhou 450003, China; yinhj410@126.com

[2] School of Water Resources Science and Engineering, University of Zhengzhou, Zhengzhou 450001, China; xiaodugege@yeah.net (X.D.); huanganqididi@163.com (A.H.)

\* Correspondence: zwg_wendy@163.com; Tel.: +86-0371-6602-0856

**Abstract:** Proper water use requires its monitoring and evaluation. An indexes system of overall water use efficiency is constructed here that covers water consumption per 10,000 yuan GDP, the coefficient of effective utilization of irrigation water, the water consumption per 10,000 yuan of industrial value added, domestic water consumption per capita of residents, and the proportion of water function zone in key rivers and lakes complying with water-quality standards and is applied to 31 provinces in China. Efficiency is first evaluated by a projection pursuit cluster model. Multidimensional efficiency data are transformed into a low-dimensional subspace, and the accelerating genetic algorithm then optimizes the projection direction, which determines the overall efficiency index. The index reveals great variety in regional water use, with Tianjin, Beijing, Hebei, and Shandong showing highest efficiency. Shanxi, Liaoning, Shanghai, Zhejiang, Henan, Shanxi, and Gansu also use water with high efficiency. Medium efficiency occurs in Inner Mongolia, Jilin, Heilongjiang, Jiangsu, Hainan, Qinghai, Ningxia, and Low efficiency is found for Anhui, Fujian, Jiangxi, Hubei, Hunan, Guangdong, Guangxi, Chongqing, Sichuan, Guizhou, Yunnan, and Xinjiang. Tibet is the least efficient. The optimal projection direction is $a* = (0.3533, 0.7014, 0.4538, 0.3315, 0.1217)$, and the degree of influence of agricultural irrigation efficiency, water consumption per industrial profit, water used per gross domestic product (GDP), domestic water consumption per capita of residents, and environmental water quality on the result has decreased in turn. This may aid decision making to improve overall water use efficiency across China.

**Keywords:** accelerating genetic algorithm; evaluation index; projection pursuit; water use efficiency; water management

## 1. Introduction

In 2013, China's total water resources were more than 2.77 trillion $m^3$, which ranked fifth in the world; however, per capita use was only 2100 $m^3$, which was just 28% of the world average. It was estimated that of all 600 cities in China, more than 400 were short of water and more than 200 suffered serious water shortages [1]. Despite the lack of water, wasting of water and inefficient water use are common and are inevitable consequences of the country's rapid, widespread but limitedly sustainable economic development. Therefore, gradually increasing the efficiency of water use is a core part of helping to bring about sustainable economic and social development, given the limited total water resources. Better water use, both domestically and industrially, and also better management of water in the environment, both in qualitative and quantitative terms, will bring economic, social, and ecological benefits [2]. Given the need for improved water use, it is necessary to establish a method of examining the overall efficiency of water use.

Water use efficiency has always been a core issue in different countries [3,4]. Scholars from all over the world have made a lot of attempts and research on the issue of water use efficiency. The research on agricultural water use efficiency started in the middle of the twentieth century [5]. Wallace explored how to achieve balance between cultivated land and water resources by examining the efficiency of agricultural water use [6]. Xu [7] analyzed agricultural water use efficiency, by constructing a bio-economy index, including soil productivity attenuation and land productivity. Lilienfel and Asmild [8] used the data envelopment analysis (DEA) method to analyze irrigation efficiency in 43 irrigated areas in the western United States. Andre et al. [9] evaluated agricultural water use efficiency in Spain. Varghese et al. [10] discussed the relationship between agricultural irrigation and agricultural production. In particular, the irrigation water use efficiency is significantly affected by the demand for import-export business and the endowment of water resources [11], the proportion of canal water use, water price, the technology of water-saving irrigation [12], and institutions for water utilization [13]. In industrial water use efficiency, Raskin et al. summarized the current status of water use worldwide, and applied water resources sustainability indicators to analyze the future trend of industrial water use efficiency [14]. Zhu [15] analyzed industrial water use and calculated the allocative efficiency of industrial water resources in 30 mainland Chinese province-level divisions (excluding Tibet). The results allowed the ranking of industrial water use efficiency in six geographic regions of China (North China, Northeast China, East China, Northwest China, Central and South China, Southeast China). Sun et al. [16] proposed that industrial water usage could be made more efficient through its evaluation by stochastic frontier analysis. There is no relevant research at home and abroad on urban water use efficiency and environmental water efficiency. In urban water use efficiency, Moglia et al. [17] assessed urban water use efficiency by a multi-standard decision-making assessment using subjective logic. Based on the comprehensive analysis of the characteristics of urban domestic water use, Fang et al. [18] proposed the concept of improving the efficiency of domestic water use from the perspective of throttling. Lee et al. [19] analyzed the effects of various water efficiency appliances on long-term water saving and water use trends. Oriana et al. [20] evaluated the metabolism-based performance of a number of centralized and decentralized water reuse strategies and their impact on integrated urban water systems (UWS) based on the nexus of water-energy-pollution. In recent years, global researches on the efficiency of water use in the ecological environment have gradually increased. Ma et al. [21] studied the strategies of agricultural water users and local governments in the protection and management of ecological water use in Xinjiang. Jiang et al. [22] reviewed the current status of rainwater harvesting systems in the Loess Plateau of China and analyzed the water use efficiency of various rainwater harvesting technologies, providing new ideas for water resources utilization research.

The literatures concerning water use efficiency in China and elsewhere mostly consider a single aspect of water use (e.g., industry or agriculture). Comprehensive studies of all aspects of water use are rare and generally only consider specific regions, rather than a whole country. There is also a lack of analysis and comparison between regions and sectors. This study seeks to provide the foundations for a thorough understanding of water use at the national and regional levels and across different sectors. Agricultural, industrial, urban, and environmental water use efficiency are considered, as is water use efficiency for each province in China and the whole country. The analysis employs the projection pursuit cluster method and the accelerating genetic algorithm [23,24]. The results allow a comparison of overall water use efficiency in 31 mainland Chinese provinces.

## 2. Evaluation Index System and Analysis of Water Use Efficiency in China

Water use industries include industrial, agricultural, and urban/domestic use, and environmental water-quality is also an important issue. Any comprehensive assessment of water use should consider all of these factors. Therefore, five water use efficiency evaluation indexes are chosen: an index concerning water use per economic output; an index for each of agricultural, industrial, urban water use efficiency; and an index for environmental water quality, as shown in Figure 1.

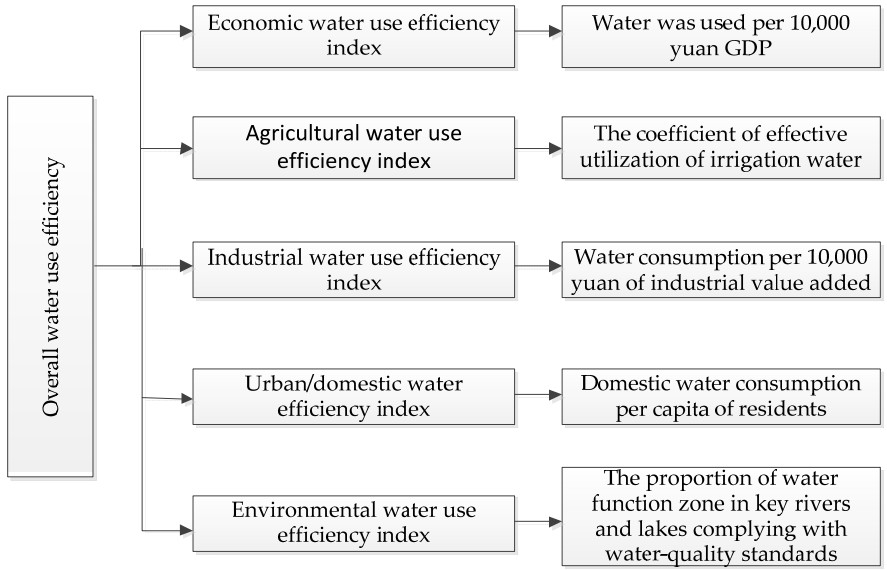

**Figure 1.** Water use efficiency assessment indexes system.

*2.1. Economic Water Use Efficiency Index*

The economic water use efficiency considers water consumption per 10,000 yuan (CNY) GDP. Based on various industries, natural geographical conditions and socio-economic development, the calculation can indirectly reflect the water use efficiency of the construction industry and the tertiary industry. The calculation Equation is as follows:

$$Q_y = \frac{Q_Z}{B_Z} \times 10000 \tag{1}$$

where, $Q_y$ is the water use per 10,000 yuan of GDP, and the unit is m$^3$/10,000 yuan. $Q_z$ is the total water use of all industries, and the unit is 100 million m$^3$. $B_z$ is the total GDP of all industries, and the unit is 100 million yuan.

In 2013, 109 m$^3$ water was used per 10,000 yuan GDP (at 2013 prices) in China. Broken down by province (Figure 2), the water use ranged from 17 to 703 m$^3$ water per 10,000 yuan GDP; Xinjiang had the highest value and Tianjin the lowest. Beijing was another notable case with 19 m$^3$ per 10,000 yuan GDP. Eight areas (including Beijing, Tianjin, Shandong, and Liaoning) had water use below 60 m$^3$ per 10,000 yuan GDP. For international comparison, Japan used 168 m$^3$ water per 10,000 United States dollars (USD) GDP in 2013 [25], and the United States of America used 358 m$^3$ per 10,000 USD GDP in 2013 [26]. The Chinese equivalent was 569 m$^3$ per 10,000 USD GDP in 2013.

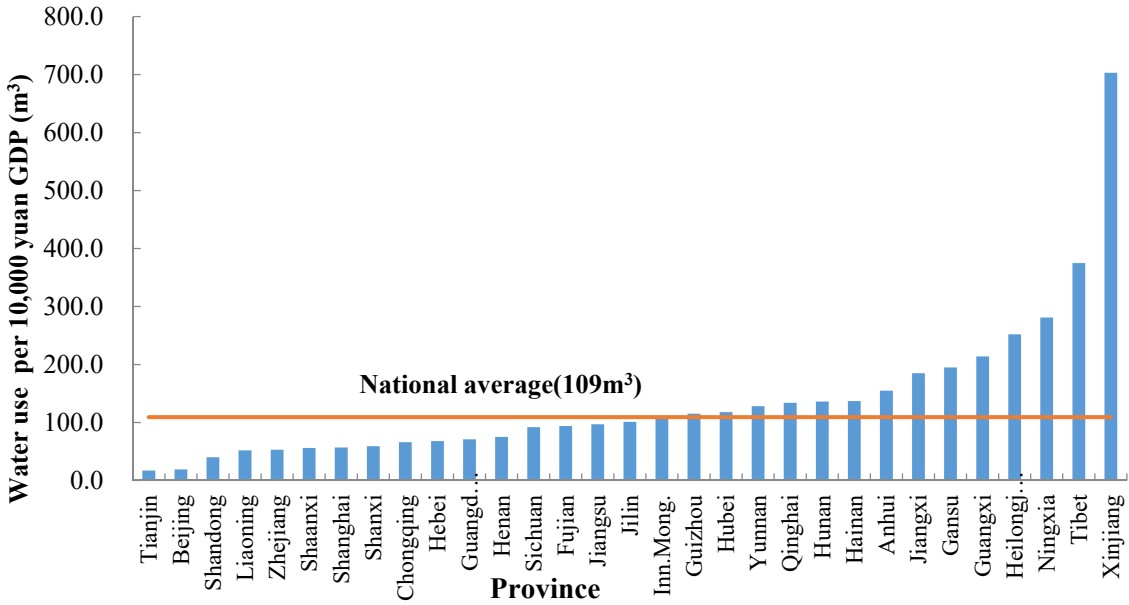

**Figure 2.** Water use per 10,000 yuan of GDP of Chinese provinces in 2013.

## 2.2. Agricultural Water Use Efficiency Index

As mentioned before, the agricultural index considers coefficient of effective utilization of irrigation water. It is the ratio of the amount of irrigation water that can be absorbed and utilized by the crop in the field to the total amount of irrigation water used by the irrigation system. Its calculation method is as follows.

$$\eta_w = \frac{W_j}{W_a} \quad (2)$$

where, $\eta_w$ is coefficient of effective utilization of irrigation water; $W_j$ is total net irrigation water in irrigation area (m$^3$); $W_a$ is total gross irrigation water use in irrigation area (m$^3$). $W_j$ is calculated by multiplying $M$ (the average irrigation water use amount per km$^2$) by the actual irrigation area A of the irrigation area. $M$ is obtained from the following Equation (3):

$$M = \frac{\sum_i^N M_i A_i}{A} \quad (3)$$

where $M_i$ is the average irrigation water use amount per km$^2$ of crop $i$; $A_i$ is the planting area of crop $i$; $N$ is the number of crop species; $A$ is the actual irrigated area in the irrigated area, $A = \sum_i^N A_i$. The average amount of irrigation water of per crop is obtained by selecting typical fields in the irrigation area and then calculating by direct measurement and observation analysis.

In 2013, the Chinese national average coefficient of effective utilization of irrigation water was 0.523. Across the 31 provinces, the coefficients ranged from 0.404 to 0.727, with the highest in Shanghai and the lowest in Tibet (Figure 3). As an international comparison, Israel's coefficient of effective utilization of irrigation water was 0.73–0.8 in 2013 [27].

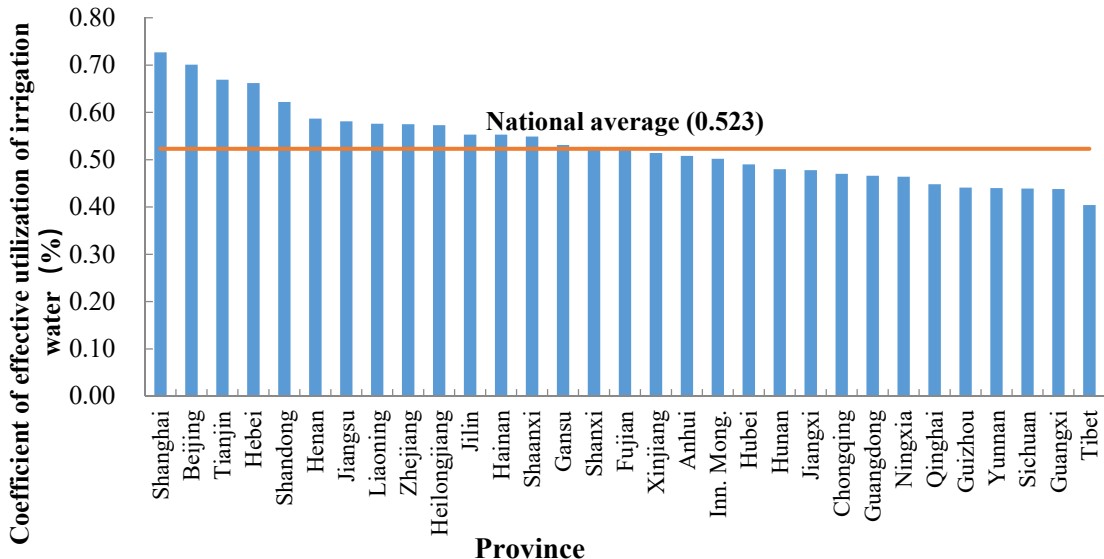

**Figure 3.** Coefficient of effective utilization of irrigation water of Chinese provinces in 2013.

Irrigation water was most effectively used in Shanghai, Beijing, and Tianjin, owing to their intensive land use with widespread use of water-saving irrigation projects. Irrigation using wells was most common in Hebei, Shandong, and Henan in North China, where water resources are deficient and require careful management; their coefficients were 0.662, 0.622, and 0.587, respectively, higher than for other provinces (but below the above-mentioned three urban municipalities). Poor infrastructure and outdated technology led Tibet to have the least-efficient irrigation, with a coefficient of 0.404.

Of the 31 studied areas, 12 (i.e., 38.7%) had coefficients above 0.55, 13 (i.e., 41.9%) had coefficients of 0.45–0.55, and six had coefficients of 0.35–0.45 (i.e., 19.4%).

### 2.3. Industrial Water Use Efficiency Index

In China, industrial water consumption refers to the water consumed by industrial and mining enterprises in the production process for manufacturing, processing, cooling, air conditioning, purification, washing, etc., calculated based on the new water withdrawal, not including the internal reuse of water. The measures to enhance industrial water use efficiency are to reduce consumption and improve water reuse rate in industrial and mining enterprises. The calculation Equation of water consumption per 10,000 yuan of industrial value added is as follows:

$$B = \frac{W_m}{B_m} \times 10000 \tag{4}$$

where $W_m$ is the water consumption for industrial added value, and the unit is 100 million m$^3$; $B_m$ is the industrial added value of the year, and the unit is 10,000 yuan.

In 2013, China consumed 67 m$^3$ water for every 10,000 yuan of industrial value added (at 2013 prices). Among the 31 provinces, the values varied between 8 and 272 m$^3$ (Figure 4), with the highest in Tibet. Tianjin showed the lowest value, and Shandong was next with 12 m$^3$. Four other areas also had consumptions below 20 m$^3$, Beijing, Liaoning, Shanxi, and Hebei.

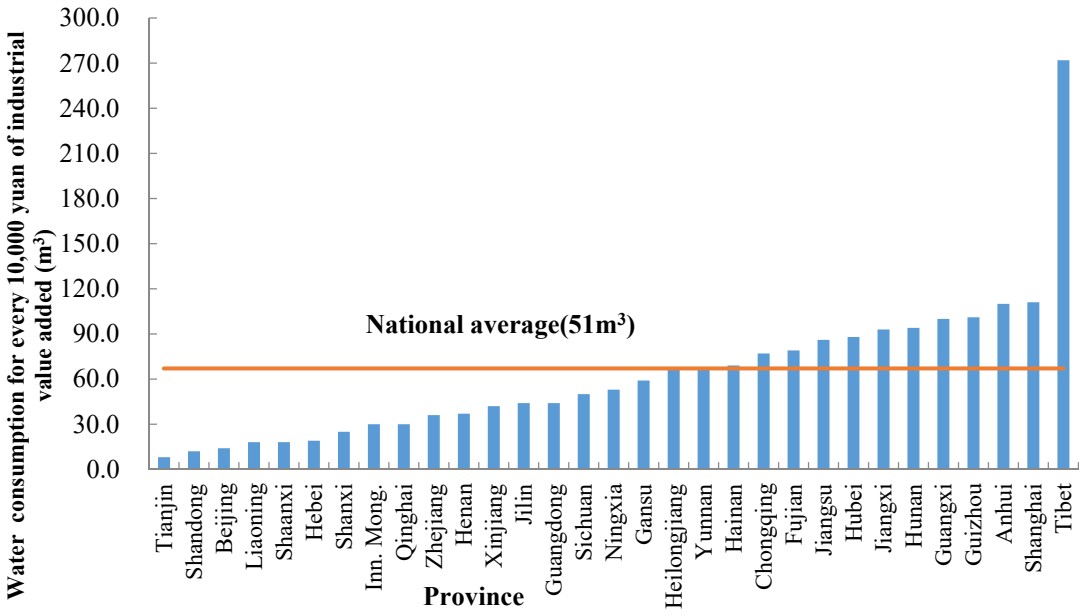

**Figure 4.** Water consumption for every 10,000 yuan of industrial value added of Chinese provinces in 2013.

### 2.4. Urban/Domestic Water Use Efficiency Index

There is a significant gap in domestic water consumption per capita among residents in different regions. The domestic water consumption per capita is a representative indicator of the efficiency of domestic water consumption to reflect the amount of water used by individuals and households in the year. The per capita domestic water consumption of residents can be calculated by the following Equation:

$$Q_u = \frac{Q_y}{p} \tag{5}$$

where $Q_u$ is the domestic water consumption per capita of residents in different areas; $Q_y$ is the total domestic water consumption of residents in the area; $P$ is the total population of the local residents.

In 2013, the average domestic water consumption per capita of Chinese residents was 38 m$^3$, and the domestic water consumption per capita of each region was unbalanced. The domestic water consumption per capita of Ningxia and Gansu residents was less than 20 m$^3$, while that of Guangdong was 63 m$^3$, and that of Guangxi, Shanghai and Hainan was more than 50 m$^3$. The domestic water consumption per capita of residents in these regions was significantly higher than the average level of China (Figure 5).

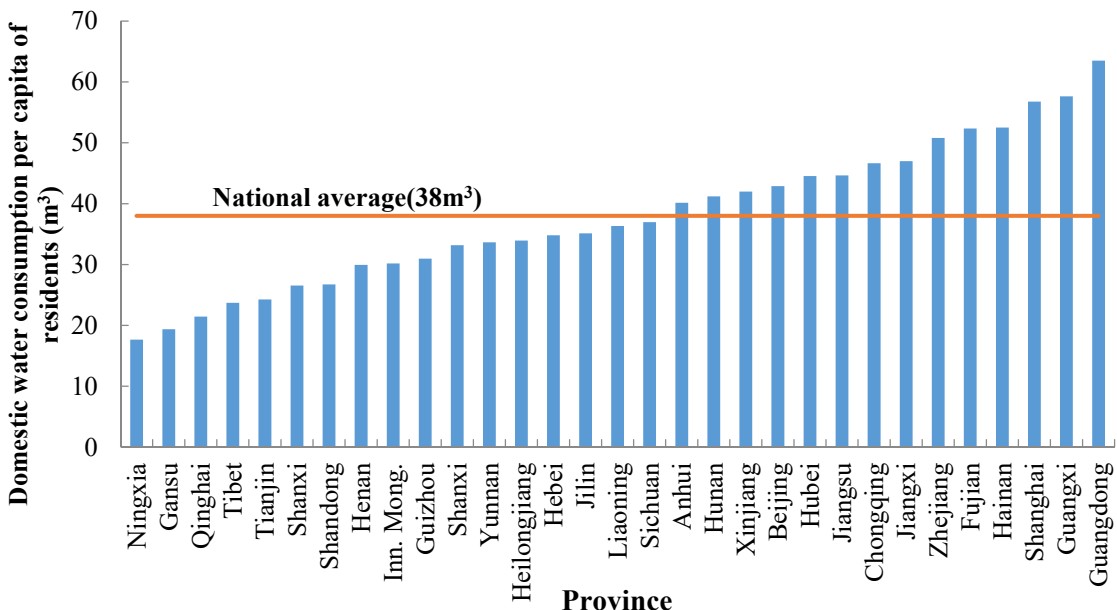

**Figure 5.** Domestic water consumption per capita of residents in 2013.

*2.5. Environmental Water Use Efficiency Index*

The main pollutants monitored by the water quality compliance rate in the water function zones are the $COD_{Mn}$ index and the $NH_3$-N index which are monitored monthly in key rivers and lakes in each province. The standard water function areas can be calculated with the following Equation:

$$S = \frac{s_i}{12} \tag{6}$$

where $S$ is the standard water function zone (water quality compliance rate is greater than 80%); $s_i$ is the number of times that the $COD_{Mn}$ index and $NH_3$-N index monitored monthly in a single water function zone both reach the standard; 12 is the total number of the times that the $COD_{Mn}$ index and $NH_3$-N index are monitored in a single water function zone.

The ratio of the water function zones in key rivers and lakes meeting the water quality standards in each province to the total water function zones in key rivers and lakes monitored is the proportion of water function zone in key rivers and lakes complying with water quality standards in China, calculated by the following Equation:

$$P = \frac{N}{M} \tag{7}$$

where $N$ is the number of the water function zones in key rivers and lakes in each province reaching the standard; $M$ is the total number of the water function zones in key rivers and lakes in each province.

In 2013, the proportion of water function zone in key rivers and lakes complying with water-quality standards in China was 67.5%. Values in the different provinces ranged between 33% and 100%, with the highest in Hainan province and the lowest in Tianjin (Figure 6).

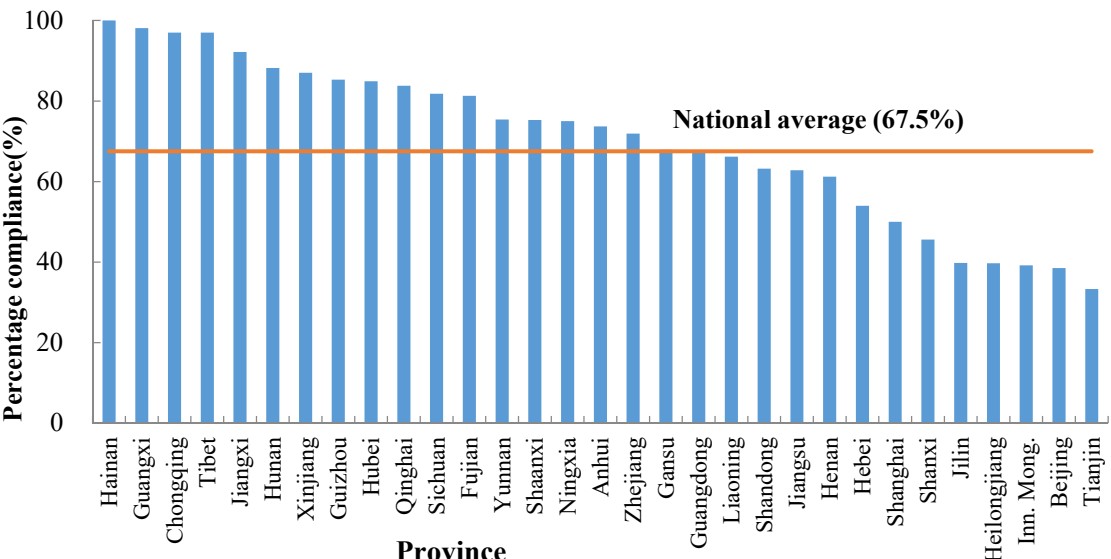

**Figure 6.** Water quality of Chinese provinces, showing the proportion of water function zone in key rivers and lakes complying with environmental water-quality standards.

## 3. Modeling the Comprehensive Index of Water Utilization Efficiency

The evaluation of overall water utilization efficiency involves selecting appropriate indicators and combining them in a system using an appropriate mathematical method. The methodology can then be applied to evaluate water use in any region, city, or river basin. This article evaluates water use efficiency based on the projection pursuit cluster method and accelerating genetic algorithm model. The model includes the following four steps.

(1)  Make each evaluation index dimensionless

Let each index value of the sample set (the evaluation object set) be *i*, among which {$x^*(i, j)$} has the positions *i* and *j* as a sample and an index value, which represent the number of samples (sample size) and the number of indicators, respectively. The extreme value normalization method is used to eliminate the dimension of each index and unify each index value change range, as follows:

When one index belongs to benefit index (that is, the index of the bigger the better), the extreme value normalization method is such as below:

$$x(i, j) = \frac{x^*(i, j) - x_{\min}(j)}{x_{\max}(j) - x_{\min}(j)} \tag{8}$$

When one index belongs to cost index (that is, the index of the smaller the better), the extreme value normalization method is such as below:

$$x(i, j) = \frac{x_{\max}(j) - x^*(i, j)}{x_{\max}(j) - x_{\min}(j)} \tag{9}$$

where $x_{\max}(j)$ and $x_{\min}(j)$ are respectively the maximum and minimum *j* index values from the sample set.

(2)  Construct the projection index function

The projection pursuit method puts *p* dimensional data {$x(i,j)|j = 1\sim p$} integrated in $a = (a(1), a(2), \ldots, a(p))$ as a projection value of single dimension based on the projection direction $z(i)$:

$$z(i) = \sum_{j=1}^{p} a(j)x(i, j), \quad i = 1, 2, \cdots, n \tag{10}$$

The according sequence $\{z(i)|i = 1\sim n\}$ can be classified using a one-dimensional scatter diagram. The term $a$ in Equation (10) is the unit length vector. That is,

$$\sum_{j=1}^{p} a^2(j) = 1 \tag{11}$$

For the synthetic projection value, the spread characteristic of the projection value $z(i)$ must be as follows. The local projection point should be as crowded as possible, with several points condensed into groups. However, the groups should spread out as much as possible as a whole. On this basis, the projection index function can be constructed as

$$Q(a) = S_z D_z \tag{12}$$

where $S_z$ is the projection value of standard deviation $z(i)$, and $D_z$ is the projection value of local density $z(i)$. That is,

$$S_z = \frac{\sum_{i=1}^{n} (z(i) - \bar{z})^2}{n - 1} \tag{13}$$

$$D_z = \sum_{i=1}^{n} \sum_{j=1}^{p} (R - r_{ij}) u(R - r_{ij}) \tag{14}$$

where $\bar{z}$ is the mean value of the sequence $\{z(i)|i = 1\sim n\}$ and $R$ is used to calculate the local density of the window radius. The value of $R$ should be large enough to allow, on average, a sufficient number of projection points in the window, to avoid making the moving average deviation too large, but it should not increase too fast with increasing $n$. A general value for $R$ is 0.1 $S_z$. The distance $r_{ij} = |z(i)-z(j)|$; and $u(t)$ is a unit speed function; when $t \geq 0$, its function value is 1, and when $t < 0$, its function value is 0.

(3)   Optimize the projection index function

When the index value of the sample set is given, the projection index function $Q(a)$ changes only with changes in the projection direction $a$. Different projection directions reflect different data structure features.

The best projection direction is that which can characterize a certain type of feature structure in high-dimensional data as much as possible. The optimal projection direction can be estimated by maximizing the projection index function:

$$\max Q(a) = S_z D_z \tag{15}$$

$$s.t. \quad \sum_{j=1}^{p} a^2 (j) = 1, \quad a(j) \in [0, 1] \tag{16}$$

where $\{a(j)|j = 1\sim p\}$ is the optimization variable used to solve the complicated non-linear problem, which is difficult to solve using conventional optimization methods. The real coding based on Accelerating Genetic Algorithm (RAGA) can simply and effectively solve the above problems via simulating the evolution process of the superior winning subjects in nature and chromosome exchange theory.

Principle of the RAGA and its implementation flow are such as below.

The RAGA has parallel processing selection, crossover, and mutation, giving it an extensive actual search scope and wide opportunity to obtain the best overall solution. The precision of its numerical solutions can be enhanced by increasing the circulation times via gradual adjustment and reducing the searching space of the variable to be optimized.

The choosing, crossing, and variation operations of the RAGA produce three generation groups; next, a choice of $N$ (group size) excellent individuals is made for the next parent population. After operation for a certain time, the accelerating genetic algorithm is started, and the selective areas for the excellent individuals are reduced (respectively, $S$ excellent individuals from $M$ times evolutionary iterations, in total a variation interval of $M \times S$ is used as the next variable interval of the accelerating genetic algorithm). In this way, the evolutionary iteration and the accelerating genetic algorithm alternately and repeatedly take the genetic evaluative gradually closer to the optimal individual. As the proximity to the optimal individual and the density of individuals increase, the probability of premature convergence is reduced to some extent. The process of the accelerating genetic algorithm is shown in Figure 7.

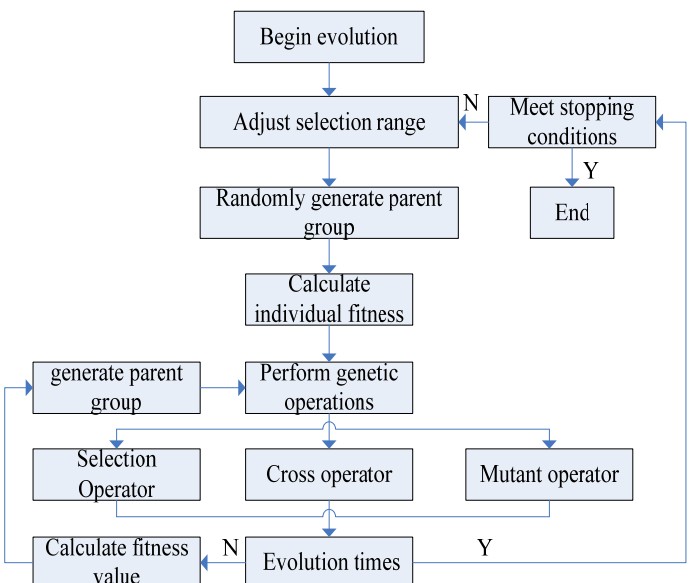

**Figure 7.** Principle of the RAGA and its implementation flow.

(4)    Cluster (arrange in the proper order)

Putting the best projection direction result $a^*$ from Equation (12) into Equation (10) gives the projection value $z^*(i)$ for each sample point. Subsequent sorting by $z^*(i)$ gives a comprehensive evaluation order based on the utilization efficiency of water resources in the different regions.

## 4. Results and Discussion

### 4.1. Results

Using the example of water use in China's 31 province-level divisions in 2013, we synthetically evaluate the water use efficiency based on the projection pursuit cluster model and the RAGA. First, the index values for each province are normalized. These indexes are economic water use efficiency, agricultural water use efficiency, industrial water use efficiency, urban water distribution efficiency, and environmental water quality. A projection index function then emerges by combining the sample sets using the substitution Equations (10), (12)–(14), The maximum projection index function is 0.7014, and the optimal projection direction is $a^* = (0.3533, 0.7014, 0.4538, 0.3315, 0.1217)$. According to Equation (11), the weight of each indicator (the contribution of the indicator to the result) is determined as $(a^*)^2 = (0.1248, 0.4920, 0.2100, 0.1099, 0.0148)$. These values are obtained according to the problems identified by Equations (15) and (16) and optimized with the RAGA. The projected values for each province are then acquired by substituting $a^*$ into Equation (15). The results are shown in Figure 8.

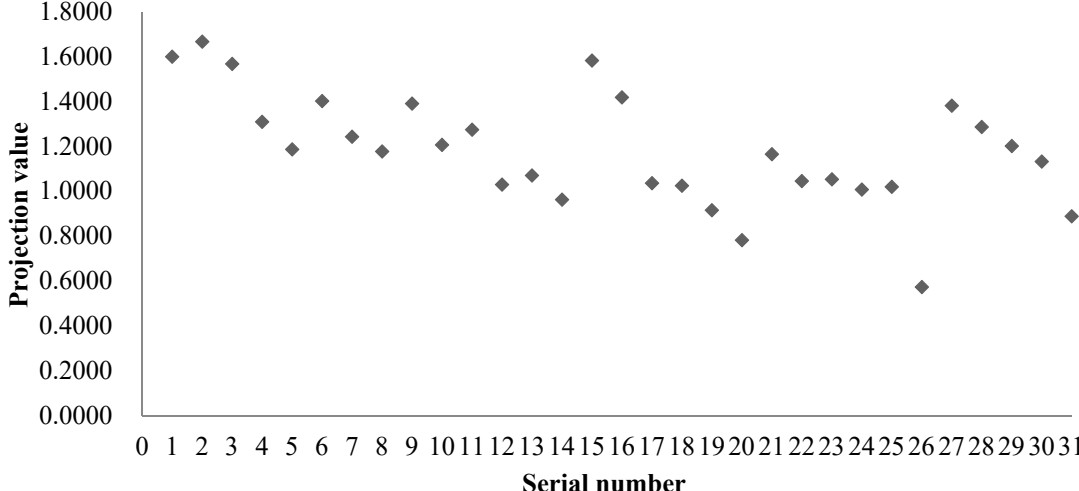

**Figure 8.** Scatter diagram of projection values for the comprehensive evaluation of water use efficiency by province.

The projection values (and hence water use efficiency) given in Table 1 and Figure 8 are ranked from highest to lowest as follows: 2, 1, 15, 3, 16, 6, 9, 27, 4, 28, 11, 7, 10, 29, 5, 8, 21, 30, 13, 23, 22,17, 12, 18, 25, 24,14, 19, 31, 20, 26. Area 2 (Tianjin) shows the highest efficient water use, while area 26 (Tibet) shows the lowest.

**Table 1.** Projection values for the comprehensive evaluation of water use efficiency.

| Sample | Serial Number | Projection Value | Sample | Serial Number | Projection Value |
|---|---|---|---|---|---|
| Beijing | 1 | 1.5993 | Hubei | 17 | 1.0355 |
| Tianjin | 2 | 1.6664 | Hunan | 18 | 1.0245 |
| Hebei | 3 | 1.5675 | Guangdong | 19 | 0.9148 |
| Shanxi | 4 | 1.3088 | Guangxi | 20 | 0.7820 |
| Inner Mongolia | 5 | 1.1866 | Hainan | 21 | 1.1651 |
| Liaoning | 6 | 1.4020 | Chongqing | 22 | 1.0447 |
| Jilin | 7 | 1.2427 | Sichuan | 23 | 1.0527 |
| Heilongjiang | 8 | 1.1770 | Guizhou | 24 | 1.0073 |
| Shanghai | 9 | 1.3900 | Yunnan | 25 | 1.0193 |
| Jiangsu | 10 | 1.2063 | Tibet | 26 | 0.5730 |
| Zhejiang | 11 | 1.2740 | Shanxi | 27 | 1.3807 |
| Anhui | 12 | 1.0291 | Gansu | 28 | 1.2859 |
| Fujian | 13 | 1.0700 | Qinghai | 29 | 1.2010 |
| Jiangxi | 14 | 0.9620 | Ningxia | 30 | 1.1317 |
| Shandong | 15 | 1.5823 | Xinjiang | 31 | 0.8879 |
| Henan | 16 | 1.4184 | | | |

*4.2. Discussion*

4.2.1. Analysis on the Overall Efficiency of Water Use in Each Province

The provinces with the highest water use efficiency are concentrated in the Beijing-Tianjin-Hebei region. The provinces with lower and lowest water use efficiency are located in the marginal regions of China such as Xinjiang, Heilongjiang, Guangxi, Yunnan, and Tibet. The overall water use efficiency of China shows a trend of gradual decline from the Beijing-Tianjin-Hebei region as the center to the surrounding areas, as shown in Figure 9. The water use efficiency is affected by the total amount of water resources and the status of regional economic development. The water use efficiency of different industries is different in different regions. The present situation of the total water resources distribution

in China also presents the trend of more in the south and less in the north as well as more in the east and less in the west. Most of the water resources in the Beijing-Tianjin-Hebei region are drawn from the mid-line project of the South-to-North Water Transfer Project in China, which determines that the total amount of water resources available in Beijing, Tianjin, and Hebei is small. Under the limitation of total amount of water resources, the economically developed areas such as Beijing, Tianjin, and Hebei must improve the water use efficiency. The water saving and water use efficiency policies formulated by the Chinese government are first carried out in the economically developed areas such as Beijing, Tianjin, and Hebei, which speeds up the pace of improving the efficiency of water resource utilization. At the same time, the industrial and agricultural technology development in Beijing, Tianjin, and Hebei is also the basis for efficient utilization of water resources. On the contrary, economic development in Xinjiang, Heilongjiang, Guangxi, Yunnan, and other regions is backward; the awareness of efficient use of water resources is not high, and the total amount of water resources is relatively abundant, resulting in low water use efficiency of these provinces. This study makes a detailed description of China's current water use efficiency, gives clear data support for the Chinese government to formulate guidelines for water use management, and provides decision-making basis for China to further improve the comprehensive level of water use efficiency.

4.2.2. Analysis on the Overall Efficiency of Water Use in Each Province

According to the best projection direction, the impact degree of each evaluation index on the evaluation results can be further analyzed. The best projection value of each index shows that the impact degree of the evaluation index on the water use efficiency of each province decreases in turn, such as the coefficient of effective utilization of irrigation water, the water consumption per 10,000 yuan of industrial added value, the water use per 10,000 yuan of GDP, domestic water consumption per capita of residents, and the water quality compliance rate of water function zones. The effective utilization coefficient of irrigation water in middle farmland reached 0.7104. China is a big agricultural country, and agricultural water use accounts for more than 60% of the country's total water use. There is also a gap in agricultural water use efficiency among different regions. The proportion of agricultural water use in economically developed areas such as Beijing, Tianjin, Hebei, and Shanghai is relatively small. Moreover, the coefficient of effective utilization of irrigation water is above 0.65, while the irrigation coefficient of farmland in Henan, Anhui, Guangxi, and Jiangxi provinces is only about 0.5. In the areas with high water use efficiency, the coefficient of effective utilization of irrigation water, the water use per 10,000 yuan of GDP, and water consumption per 10,000 yuan of industrial value added are ranked first. Domestic water consumption per capita of residents is also above the provincial average. The index value of the proportion of water function zone in key rivers and lakes complying with water-quality standards is ranked lower. In the areas with low water use efficiency, the first four indicators are ranked very low. From the final evaluation results, the proportion of water function zone in key rivers and lakes complying with water-quality standards has little influence on the evaluation results. Therefore, the improvement of water use efficiency should focus on Economic water use efficiency, agricultural water use efficiency, and industrial water use efficiency, taking into account the efficiency of domestic and ecological water use. Through comprehensive analysis of various indicators, it is possible to clarify the water use efficiency of various provinces and industries and help local governments improve water use efficiency in different provinces and industries, thereby improving overall water use efficiency.

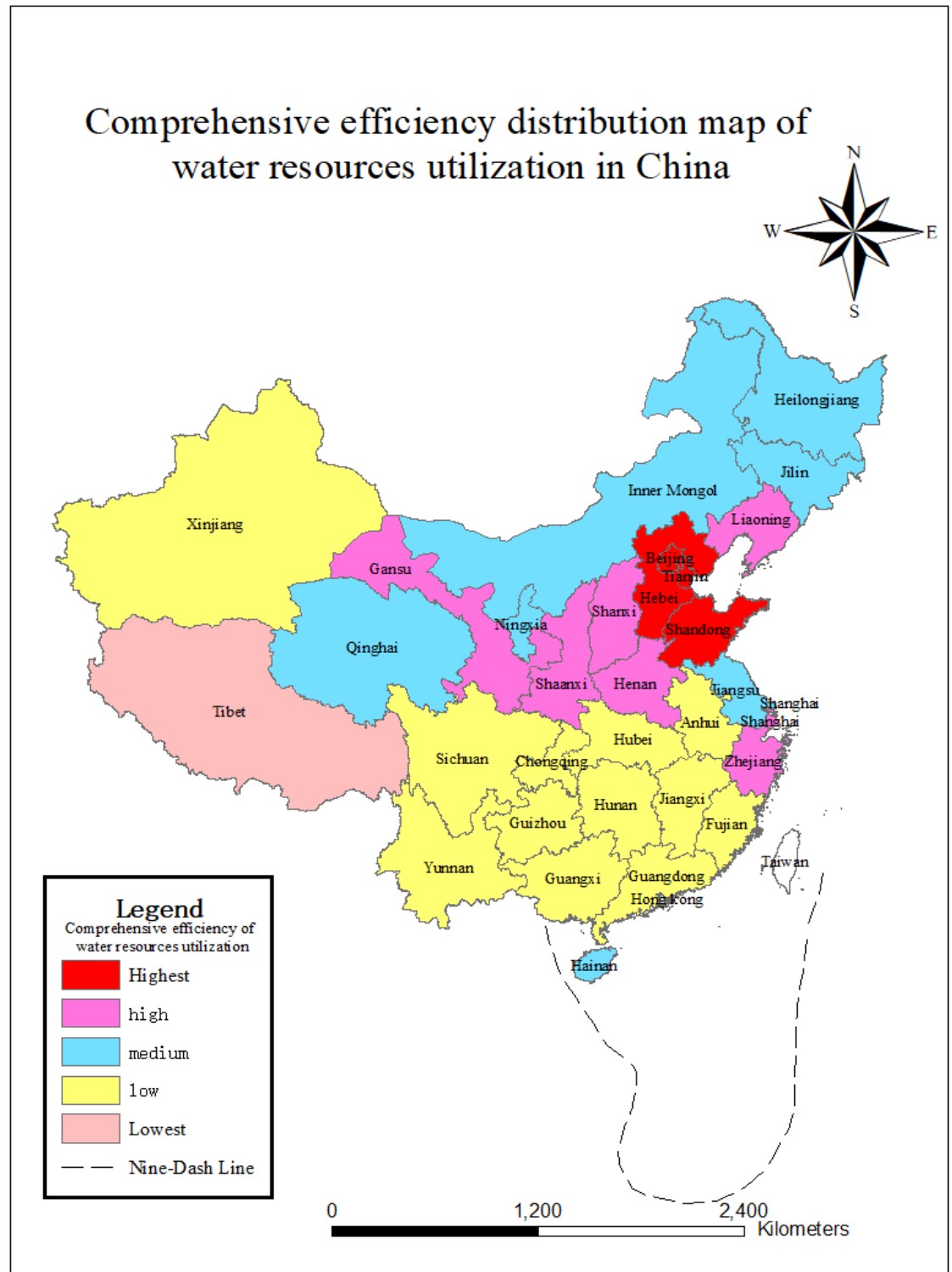

**Figure 9.** Overall water use efficiency across mainland China. (Note that consistent data for Hong Kong, Macao, and Taiwan are unavailable, and thus these regions are excluded from the analysis).

4.2.3. Analysis on the Overall Efficiency of Water Use in Each Province

From the calculation results, the overall level of China's water use efficiency is low, and only a few provinces reach higher and the highest levels; therefore, there is still huge room for improvement in China's water use efficiency. Improving various industries' water use efficiency is of great significance to China. In China, to improve the efficiency of water use, we should first improve the efficiency of

agricultural water use. We can gradually realize the efficient utilization of agricultural water resources by means of water saving measures such as trimming irrigation canal system, increasing the coefficient of water utilization of canal system, and expanding the irrigation area of sprinkler irrigation and drip irrigation. There is a clear gap between the provinces and the water consumption per 10,000 yuan of industrial added value. The water consumption of Tianjin's industrial added value is only 8 m$^3$, while Tibet has reached 272 m$^3$, and Guizhou Province has reached 101 m$^3$. Therefore, for industrially underdeveloped areas such as Tibet, Guizhou, and Yunnan, water saving measures such as upgrading technology, upgrading equipment, and adjusting industrial structure can also greatly improve water use efficiency. The government should play a leading role in the process of water use, strictly control the use of water resources, and coordinate the use of water resources among various water departments.

## 5. Conclusions

This study has established an index to assess the overall water use efficiency, based on the projection pursuit cluster and accelerating genetic algorithm method. The index was applied to study the overall water use efficiency in mainland China's 31 province-level divisions. The calculation and analysis led to the following three main findings.

(1) The evaluation index of overall water-use efficiency covers all areas of water use, including all industries. This broad analysis and regional comparison across all provinces provide a sound understanding of the effectiveness of water management at the national, regional, and sector levels.

(2) This paper gives the detailed steps of the combined projection pursuit cluster and accelerating genetic algorithm method. The method is applied to an analysis of water use in China. Each province is ranked according to its overall water-use efficiency, and contribution of each index to the overall result can be reflected by the optimized projection direction. This method is easily applicable, gives accurate and objective evaluation results, and demonstrates projection pursuit as a powerful tool for a variety of comprehensive evaluation applications.

(3) The projection pursuit cluster model can be mapped to low-dimensional subspace by finding the best projection direction, which is advantageous to the aggregation process and the comprehensive assessment of the sample. A shortcoming of this model is that the best projection direction is dependent on the optimization algorithm and is computationally expensive to calculate.

**Author Contributions:** Conceptualization, W.Z. and X.D.; methodology, W.Z. and X.D.; validation, W.Z., H.Y. and A.H.; formal analysis, W.Z., X.D and H.Y.; data curation, X.D.; writing—Original draft preparation, W.Z. and X.D.; writing—review and editing, H.Y.; supervision, A.H.; project administration, W.Z.; funding acquisition, W.Z.

**Funding:** This research was funded by the National Natural Science Foundation of China (51979119).

**Acknowledgments:** The authors are very grateful to the editors and reviewers for their support and help.

**Conflicts of Interest:** The authors declare no conflicts of interest.

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
