# Peer review of "Analysis and Comprehensive Evaluation of Water Use Efficiency in China"

_water, doi:10.3390/w11122620_

Round 1
Reviewer 1 Report
please see attachment

Reviewer 2 Report
1. It is not appropriate to analyze the water efficiency of industrial structure without considering it.
2. Several basic data in lines 31-33 need to be added and added with references.
3, line 71, not only a document needs to use plural
4, 2013 data is relatively old, it is recommended to update to 2017 or 2018
5, the unit proposal of Figure 3 is changed to an integer
6, Figure 8 is not standardized, it is recommended to add scale, north arrow and nine-section line, the provincial boundary line and the national boundary line are filled with color
7, please delete the number "22" of line 319
8, the research method is relatively simple, the method is less innovative
9, there are some problems in the English language expression, please touch up
Reviewer 3 Report
This manuscript constructed an index to assess the overall efficiency of water use, based on the projection pursuit cluster and accelerating genetic algorithm method. The 31 provinces in mainland China were chosen for case study. Some interesting results were obtained. However, the manuscript lacks detailed explanation for the evaluation indicators and methods selection. Further analysis and discussion are needed for the evaluation results of water efficiency in each province. I would suggest a major revision for this article before it can be accepted for publication. Below are some detailed comments and suggestions:
Line 12: The indicators that constitute the index of overall water-use efficiency need to be more specific. Please add which indicators are considered among agricultural, industrial, urban water-use efficiency, and environmental water efficiency.
Line 21-23: Specify the contribution of each evaluation index for the index of overall water-use efficiency, since it has been analyzed in the discussion section.
Line 83: The relationship between water environment quality and water use needs to be demonstrated more specific.
Line 87: Why choose the economic water-use efficiency as an index and why select the water consumption per 10,000 yuan (CNY) GDP to represent it? Please clarify.
Line 98-100: The comparison of data between countries in different years is inappropriate. It's better to find data in the same year for comparison.
Line 110: ‘Total irrigation water use;’ should be deleted.
Line 140-141: unify the expression of ‘environmental water efficiency index’ and ‘water‐environmental quality index’ maybe more proper.
Figure 8: The Chinese map lacks nine segments, and the boundaries of the provinces in the map are unclear. Besides, the classification standards of water-use efficiency evaluation results in the map need to be explained.
Line 243: a* is the optimal projection direction estimated with accelerating genetic algorithm previously. So it is not appropriate to use a* to reflect the contribution of the component indexes to the evaluation results. And how a* affects the evaluation results need to be discussed and analyzed.
The manuscript makes no mention about why the assessment of water-use efficiency in different provinces has such a result? It is suggested to include the reason analysis in discussion or conclusion sections of the manuscript.
The analysis of previous studies is a little lacking, and some of the literature is not suitable for the research topic of this manuscript. Please add more new literatures such as the researches in recent years by Sun Yat-sen University water resources team. The reasons why select the study methods should be explained clearly.
Round 2
Reviewer 1 Report
please see attachment

Reviewer 2 Report
The article has been greatly improved and can be accepted
Author Response
Thank you for your approval of our revised manuscript.
Reviewer 3 Report
The author have revised the manuscript and it’s appropriate to be accepted.
Author Response

(The authors gave the same response as above.)
